# SELF-ATTENTIONAL CREDIT ASSIGNMENT FOR TRANSFER IN REINFORCEMENT LEARNING

## ABSTRACT

The ability to transfer knowledge to novel environments and tasks is a sensible desiderata for general learning agents. Despite the apparent promises, transfer in RL is still an open and little exploited research area. In this paper, we take a brand-new perspective about transfer: we suggest that the ability to assign credit unveils structural invariants in the tasks that can be transferred to make RL more sample efficient. Our main contribution is SECRET, a novel approach to transfer learning for RL that uses a backward-view credit assignment mechanism based on a self-attentive architecture. Two aspects are key to its generality: it learns to assign credit as a separate offline supervised process and exclusively modifies the reward function. Consequently, it can be supplemented by transfer methods that do not modify the reward function and it can be plugged on top of any RL algorithm.

## 1 INTRODUCTION

To some, intelligence is measured as the capability of transferring knowledge to unprecedented situations. While the notion of intellect itself is hard to define, the ability to reuse learned information is a desirable trait for learning agents. The coffee test (Goertzel et al., 2012), presented as a way to assess general intelligence, suggests the task of making coffee in a completely unfamiliar kitchen. It requires a combination of advanced features (planning, control and exploration) that would make the task very difficult if not out of scope for the current state-of-the-art Reinforcement Learning (RL) agents to learn. On the other hand, it is solved trivially by humans, who exploit the universally invariant structure of coffee-making: one needs to fetch a mug, find coffee, power the coffee machine, add water and launch the brewing process by pushing the adequate buttons. Thus, to solve the coffee test, transfer learning appears necessary. Were we to possess a random kitchen simulator and a lot of compute, current transfer methods would still fall short of consistently reusing structural information about the task, hence also falling short of efficient adaptation.

Credit assignment, which in RL refers to measuring the individual contribution of actions to future rewards, is by definition about understanding the structure of the task. By structure, we mean the relations between elements of the states, actions and environment rewards. In this work, we investigate what credit assignment can bring to transfer. Encouraged by recent successes in transfer based on supervised methods, we propose to learn to assign credit through a separate supervised problem and transfer credit assignment capabilities to new environments. By doing so, we aim at recycling structural information about the underlying task.

To this end, we introduce SECRET (SElf-attentional CREdit assignment for Transfer), a transferable credit assignment mechanism consisting of a self-attentive sequence-to-sequence model whose role is to reconstruct the sequence of rewards from a trajectory of agent-environment interactions. It assigns credit for future reward proportionally to the magnitude of attention paid to past state-action pairs. SECRET can be used to incorporate structural knowledge in the reward function without modifying optimal behavior, as we show in various generalization and transfer scenarios that preserve the structure of the task.

Existing backward-view credit assignment methods (Arjona-Medina et al., 2019; Hung et al., 2018) require to add auxiliary terms to the loss function used to train agents, which can have detrimental effects to the learning process (de Bruin et al., 2018), and rely on an external memory, which hinder the generality of their approach. SECRET does neither. Also, as we show in Sec. 3.1, the architecture we consider for SECRET has interesting properties for credit assignment. We elaborate about our

novelty with respect to prior work in Sec. 4. We insist on the fact that the focus of our work is on transfer and that it is not our point to compete on credit assignment capabilities.

We would like to emphasize several aspects about the generality of SECRET: 1) our method does not require any modification to the RL algorithm used to solve the tasks considered, 2) it does not require any modification to the agent architecture either and 3) it does not alter the set of optimal policies we wish to attain. Moreover, our method for credit assignment is offline, and as a result, it can use interaction data collected by any mean (expert demonstrations, replay memories (Lin, 1992), backup agent trajectories...). We believe that this feature is of importance for real-world use cases where a high number of online interactions is unrealistic but datasets of interactions exist as a byproduct of experimentation.

**Background**   We place ourselves in the classical Markov Decision Process (MDP) formalism (Puterman, 1994). An MDP is a tuple $(S, A, \gamma, R, P)$ where $S$ is a state space, $A$ is an action space, $\gamma$ is a discount factor ($\gamma \in [0, 1)$), $R : S \times A \times S \to \mathbb{R}$ is a bounded reward function that maps state-action pairs to the expected reward for taking such an action in such a state. Note that we choose a form that includes the resulting state in the definition of the reward function over the typical $R : S \times A \to \mathbb{R}$. This is for consistency with objects defined later on. Finally, $P : S \times A \to \Delta_S$ is a Markovian transition kernel that maps state-action pairs to a probability distribution over resulting states, $\Delta_S$ denoting the simplex over $S$.

An RL agent interacts with an MDP at a given timestep $t$ by choosing an action $a_t \in A$ and receiving a resulting state $s_{t+1} \sim P(\cdot|s_t, a_t)$ and a reward $r_t = R(s_t, a_t, s_{t+1})$ from the environment. A trajectory $\tau = (s_i, a_i, r_i)_{i=1,\dots,T}$ is a set of state-action pairs and resulting rewards accumulated in an episode. A subtrajectory is a portion of trajectory that starts at the beginning of the episode. The performance of an agent is evaluated by its expected discounted cumulative reward $\mathbb{E}\left[\sum_{t=0}^{\infty} \gamma^t r_t\right]$. In a partially observable MDP (POMDP), the agent receives at each timestep $t$ an observation $o_t \sim \mathcal{O}(\cdot|s_t)$ that contains partial information about the underlying state of the environment.

## 2   SECRET: SELF-ATTENTIONAL CREDIT ASSIGNMENT FOR TRANSFER

SECRET uses previously collected trajectories from environments in a source distribution. A self-attentive sequence model is trained to predict the final reward in subtrajectories from the sequence of observation-action pairs. The distribution of attention weights from correctly predicted nonzero rewards is viewed as credit assignment. In target environments, the model gets applied to a small set of trajectories. We use the credit assigned to build a denser and more informative reward function that reflects the structure of the (PO)MDP. The case where the target distribution is identical to the source distribution (in which we use held-out environments to assess transfer) will be referred to as generalization or *in-domain transfer*, as opposed to *out-of-domain transfer* where the source and the target distributions differ.

### 2.1   SELF-ATTENTIONAL CREDIT ASSIGNMENT

**Credit assignment as offline reward prediction**   We learn to assign credit through an offline reward prediction task, based on saved trajectories of agent-environment interactions. We create a sequence-to-sequence (seq2seq) model (Sutskever et al., 2014) that takes as input the sequence of observation-action pairs and has to reconstruct the corresponding sequence of environment rewards. Being offline, the reward prediction task is learned separately from the RL task, and the reward prediction model does not share representations with the agent. This way, the representations learned for credit assignment do not affect or get mixed with the representations learned for control. Operating offline brings several advantages: one can directly interact with the replay memory of agents and even use expert demonstrations or arbitrary saved transitions as a source of supervision, which could be useful in settings where on-policy interactions are costly, such as robotics. We equip our seq2seq model with an attention mechanism (Bahdanau et al., 2015) and view the attention weights of the reward reconstruction task as our primary source of assigned credit. The motivation to do so is that the seq2seq model looks into the past to find predictive signal in order to reconstruct the reward, so observation-action pairs it attends to should be those which reduce its uncertainty about the future, in other words those that explain future reward and should be credited.

**On the use of observations**   In MDPs, environment states follow the Markov property: they summarize the history of previous interactions and are sufficient to predict the future. As such, predictive models are highly biased towards focusing on the current sequence element, which hinders credit assignment. Under that consideration, when dealing with MDPs, we turn states into observations by applying transformations that hide a certain amount of information from states and break the Markov assumption. For instance, in gridworlds with visual states, we crop the image and get a player centered image with a given window size. Doing so encourages the model to look into the past to find predictive signal, and allow us to track the relative importance given to each element to reconstruct the credit assigned. In POMDPs, this might be unnecessary depending on the amount of information shared between observations and true states.

**Self-attention for credit assignment**   Unlike other seq2seq architectures, self-attentive models like Transformers (Vaswani et al., 2017) have direct computational paths between pairs of sequence elements, due to their representations that depend on projections of all sequence elements. This feature is key to long-term credit assignment. As an illustration, consider an RL task where the terminal reward depends only on the first observation, which is drawn randomly. Predicting the reward correctly requires to remember the first observation, which would be very challenging for a recurrent architecture whose memory goes through $O(n)$ transformations, $n$ being the size of the sequence. On the other hand, a self-attentive model directly accesses the value of the initial observation, which makes credit assignment easier.

**Reward prediction architecture**   We use a Transformer decoder with a single self-attention layer (Lin et al., 2017) and a single attention head. The model input is a sequence of observation-action couples $(o_t, a_t)_{t=0,...,T}$. Each observation goes through a series of convolutional layers (for visual inputs) followed by a series of feed-forward layers. Each action representation, a one-hot vector in the discrete action case, is concatenated to the learned observation embedding. Those representations of dimensionality $d_i$ are combined with positional encoding (PE), fed to a self-attention layer and then to a position-wise feedforward layer that outputs logits for reward prediction classes. PE encodes the relative positions of sequence elements, see Appendix A for details.

Self-attention is an attention mechanism with parameterization $(W_k, W_q, W_v)$, each matrix belonging to $\mathbb{R}^{d_i \times d_k}$, that puts sequence elements in relation by computing non-linear similarity scores for all pairs of elements in the sequence. To do so, each sequence element is mapped to a query vector that is matched against keys and values obtained from the previous elements. To be consistent with the goal of assigning credit, the model should not be able to peek into the future. Thus, we restrict the computational window of each sequence element to the information stored in representations of the previous elements in the sequence and its own by applying a causal mask $M_c$ to the result of the pairwise similarity computations, assigning a value of $0$ to masked elements after the softmax.

Let $X = (x_t)_{t=0,...,T} \in \mathbb{R}^{T \times d_i}$ denote the input sequence in a matrix form, $x_t$ being the result of internal computations of the model on its $t^{\text{th}}$ input. In the same fashion, we note $Z = (z_t)_{t=0,...,T} \in \mathbb{R}^{T \times d_k}$ the sequence resulting from the application of self-attention. We then have

$$Z = \text{softmax}\left(\frac{M_c \odot (QK^T) - C(1 - M_c)}{\sqrt{d_k}}\right) V,$$

where $Q = XW_q \in \mathbb{R}^{T \times d_k}$ stores queries, $K = XW_k \in \mathbb{R}^{T \times d_k}$ keys, and $V = XW_v \in \mathbb{R}^{T \times d_k}$ values as linear projections of the input; $d_k$ stands for the dimension of the key vectors, $M_c \in \{0, 1\}^{T \times T}$ is a binary matrix that acts as a causal mask (a lower triangular matrix), $\odot$ is the Hadamard product and $C$ is a large constant ($10^9$ in practice).

Notably, the resulting observation-action representation can be viewed as a linear combination of the values of previous elements: $z_t = \sum_{i=0}^{t} \alpha_{i \leftarrow t} v_i$ where $\alpha_{\cdot \leftarrow t} = (\alpha_{i \leftarrow t})_{i=1,...,t} \propto \exp(\langle q_t, k_i \rangle / \sqrt{d_k})$. The vector $\alpha_t$ contains the normalized attention weights for the prediction at timestep $t$ and sums to $1$. Since observations contain only a portion of their initial information, the fact that the model succeeds in the prediction task indicates that it reconstructed the missing information from its past. Therefore, attention weights themselves can be viewed as a form of credit assignment, and will be used as such in what follows.

While performing regression on the rewards could also be an option, our experiments found that regression tends to converge to poor local optima. Consequently, we predict the sign of the experi-

enced rewards: $q(r) = \text{sign}(r)$ with $\text{sign}(0) = 0$. We chose the sign as the classification target for its invariance to the scale of the rewards. We use a weighted sequential cross-entropy as the loss function over the class-wise model predictions $f_{\theta,c}$, writing $\tau(o, a)$ the subtrajectory of $\tau$ ending with the observation-action couple $(o, a)$ to translate the effect of the binary mask:

$$\mathcal{L}_\theta(\tau) = -\sum_{c \in \{-1,0,1\}} \frac{w(c)}{|\tau|} \sum_{(o,a,r) \in \tau} \mathbb{1}\{q(r) = c\} \log \left( f_{\theta,c}(\tau(o, a)) \right).$$

We have found class weighting to be very important to keep the variance of prediction metrics across a variety of datasets of sampled trajectories low.

**Generating trajectories** To train SECRET, we generate a dataset of trajectories which contains a certain proportion of successful trajectories. If source environments are simple enough so that the task has sufficient chance to be solved by acting randomly, we use a random policy to generate trajectories. For more complex distributions of environments, we use a RL agent (either trained or in the learning phase) to generate trajectories. We think purely exploratory methods like Ecoffet et al. (2019) could have advantages over using an RL agent and leave the study of their use for future work.

## 2.2 LEVERAGING CREDIT VIA REWARD SHAPING

In this subsection, we explain how we use credit assignment to make learning more sample-efficient.

**Reward shaping** In RL, agents often deal with sparse rewards that make the learning process slow. Reward shaping Ng et al. (1999) is a technique that aims at densifying the reward so as to improve sample efficiency. It defines a class of reward functions that can be added to the original environment rewards without modifying the set of optimal policies. For a given MDP $M = (S, A, \gamma, R, P)$, we define a new MDP $M' = (S, A, \gamma, R', P)$ where $R' = R + F$ is the shaped reward and $F$ the shaping. The reward shaping theorem states that if there exists a function $\phi$ such that $F : (s, a, s') \to \gamma\phi(s') - \phi(s)$, then $M$ and $M'$ admit the same set of optimal policies. $\phi$ is called a potential function. With domain knowledge, one can use reward shaping to design more informative reward functions without encouraging unwanted behavior. Nevertheless, shaping rewards requires good priors for the task and the potential function must often be engineered manually.

Since SECRET weighs the contribution of observation-action pairs to future reward, we use it to derive a shaped reward that corresponds to the sum of future reward reachable from the underlying state, weighted by the attention calculated by the model. We explain the process in the following.

**Computing the potential function** We define the redistributed return $R_\tau^\leftarrow$ of a trajectory $\tau$ as:

$$R_\tau^\leftarrow(s, a) = \sum_{t=1}^{T} \mathbb{1}\{s_t = s, a_t = a\} \sum_{i=t}^{T} \alpha_{t \leftarrow i} r(s_i, a_i),$$

where $\alpha_{i \leftarrow j}$ is the attention weight on $(o_i, a_i)$ when predicting the reward $r_j$ and $s_i$ are environment states. Indeed, SECRET uses observations but we keep the states they are constructed from to compute the potential. In POMDPs, we recover an approximate state from the observation, either manually or through inference. In this work, we use a state constructed manually, see Appendix A for details.

To compute the potential function, we generate a set $D$ of trajectories like described in Sec. 2.1. Since we operate on trajectories, the same state-action pair can appear twice in a sequence and benefit from a different amount of attention, which is why we must include the first summation. In the reward shaping formalism, the potential function $\phi$ depends only on the state. To stay within its bounds, we define $\phi$ as the forwarded redistributed return. It is computed as the following estimate:

$$\hat{\phi}(s) = \frac{1}{|D|} \sum_{\tau \in D} \sum_{t=1}^{T} \mathbb{1}\{s_t^{(\tau)} = s\} R_\tau^\leftarrow(s_{t-1}^{(\tau)}, a_{t-1}^{(\tau)}).$$

Note that in practice we only redistribute individual rewards that were successfully predicted. Also, some states are generally missing from the data distribution induced by the set of trajectories used. For those states, we set to potential to 0, which results in a $-\hat{\phi}(s)$ additional reward when transitioning to those from the state $s$. As a result, it gives agents incentive to stay on the support of the data distribution unless they encounter high-reward states.

Because it relies on reward shaping, SECRET conserves optimal policies. We empirically find that agents learn faster with the resulting augmented reward function. A way to look at it is that we densify the learning signal and bias the agent towards behaviors that encourage future rewards.

### 2.3 Transferring Credit Assignment

We start by conveying intuition as to why SECRET should transfer to new environments. In fields other than RL, seq2seq models similar to that of SECRET have shown outstanding transfer capabilities (Devlin et al., 2018; Peters et al., 2018; Howard and Ruder, 2018), even in low-resource settings (Zoph et al., 2016). In transfer scenarios that preserve the structure of the MDP, the optimal finegrained control sequence can vary drastically from one environment to another. This is why credit assignment is an interesting alternative to the transfer of weights: given an underlying environment state and a specific action, their contribution to future rewards is not fundamentally altered. Such scenarios include specific changes in the state (or observation) distribution and changes to the reward function that preserve the optimal policies. These also include changes in the dynamics of the environment, and though it affects credit assignment, we show later on that SECRET adapts surprisingly well to such scenarios. Another point that motivates the use of our method for transfer is the fact that we keep the representations learned for credit assignment separate from the control representations learned by agents. Indeed, de Bruin et al. (2018) showed that RL representations were not optimal for transfer.

**Transfer setting** We argue that transfer should be considered effective when agents learn to solve target tasks efficiently because efficiency gains in the target domain compound while the cost of training in the source is fixed. Hence, we use the Total Target Time Scenario metric (Taylor and Stone, 2009) to assess transfer. Nevertheless, collecting trajectories in the source domain can be costly. We report the number of trajectories used to train SECRET in each scenario.

As before, SECRET is trained on episodes of interaction sampled from the source distribution. In each target environment, we sample multiple trajectories (see the following section for details about the policies used to generate the trajectories). We then compute the attentional potential function by calculating an estimate of the expected redistributed reward, as described in Sec. 2.2.

## 3 Experiments

In this section, we aim to answer the following questions: can SECRET improve the sample efficiency of learning for RL agents? Does it generalize and/or transfer? How does it compare to transfer baselines? Is the credit assigned by SECRET interpretable? The results of complementary experiments are presented in Appendix B.

**The Triggers environment** We introduce Triggers, an interpretable and customizable environment that we use to assess the quality of the credit inferred with our method. In Triggers, the agent is located in a two-dimensional bounded grid. Its actions consist solely of moving of one cell in one of the cardinal directions. Any action that would lead the agent outside the boundaries of the environment (as indicated by the walls in the figure) is ignored but still counted as an action taken by the agent. The goal of the agent (represented as a yellow square) is to activate all the switches (red squares) and then collect all the prizes (pink squares). Prizes are the only source of reward and give a $-1$ penalty unless all switches are activated, in which case they give a $+1$ bonus. Both prizes and switches disappear once collected.

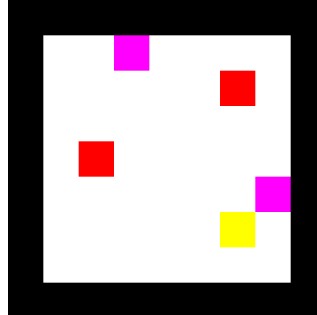

Figure 1: Example of a Triggers environment.

The main feature of Triggers is that every positive reward is conditional to the presence of a known subset of states in the agent history, and thus credit assignment can be assessed in a *rigorous* way. Some instances of Triggers can prove challenging to solve optimally for traditional RL methods since agents have to activate every Triggers before experiencing rewards. Triggers environments being MDPs, we turn their states into observations by cropping the view around the agent. We use 3x3 windows in all our experiments. Trajectories are generated with random policies.

**DMLab keys doors**   We use the `keys_doors_puzzle` 3D environment from DMLab (Beattie et al. (2016)) in which the agent must locate keys whose colors indicate the doors they open. It can only possess one key, therefore picking the wrong key prevents it from reaching further rewards. The agent receives as input what would correspond to a first person view of what is in its line of sight. It can move forward, backward and rotate. Each key picked up grants a +1 bonus, equally to each door opened. Independently, a cake rewards the agent by a +50 increase in score when collected. We do not apply any transformation to the agent inputs. In that setup, agents benefit from understanding the link between keys and doors. We hypothesized that our credit assignment mechanism might identify this relation

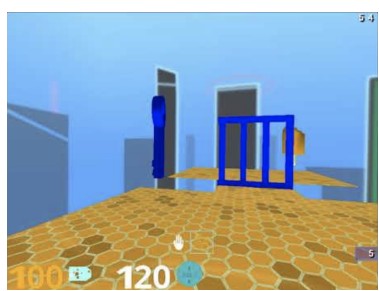

Figure 2: Observations from DMLab are first person views.

and reward the agent for picking up keys. To assert this, we modified the setting so that picking up keys does not provide rewards. Additionally, the visual input is richer than the one from Triggers environments and the average number of steps per episode is extended. Finally, agents move and rotate across the room. Since picking up a key does not require to actually see the key, it can be hard to know whether a key was taken and predict further door opening rewards. Trajectories are generated with a trained agent.

**Agents used**   We use tabular $Q$-learning (Watkins and Dayan, 1992) for in-domain and out-of-domain experiments in Triggers except for the transfer to environments with modified dynamics where we use Deep $Q$-Networks (DQN) (Mnih et al., 2015). We use Proximal Policy Optimization (PPO) (Schulman et al., 2017) agents for in-domain experiments in DMLab.

## 3.1   Credit assignment

We provide an analysis of the credit inferred by SECRET. The analysis is qualitative and quantitative, since we rely on both visual assessment and binary detection metrics.

The process of evaluating credit assignment in Triggers goes as follows: we first generate trajectories and train the model. We then compare the credit assigned by SECRET on trajectories sampled from held-out environments to a ground truth credit assignment. We build that ground truth by exploiting the exact knowledge of where triggers are. It is a vector that is 0 almost everywhere and 1 on state-action couples that precede the activation of a Triggers. By doing so, we explicitly target the state-action couples whose resulting state is causally linked to the reward experienced later.

We find the redistribution to be near optimal in simple instances of Triggers (see Fig. 3-left): attention concentrates quasi exclusively on state-action pairs that enable the collection of future reward. This is confirmed by precision-recall analysis: over the distribution of scenarios considered, a simple binarization heuristic over attention values yields an average precision of 0.96 for an average recall of 0.94. More information on the heuristic is in Appendix A.

In `keys_doors_puzzle`, we adopt the same set of experiments. Since the agent can move backward and spin, in some scenarios it takes a key that is not in is line of sight. In addition, the granularity of the state space is such that off-by-one prediction errors are common but do not hinder the credit mechanism: attributing credit to the state-action couple preceding the collection of a key or the previous one leads to imperceptible changes in the resulting shaped rewards. Fig. 3-right shows similar results as for Triggers. Appendix A also provides a heatmap for this task that shows that attention concentrates around the keys.

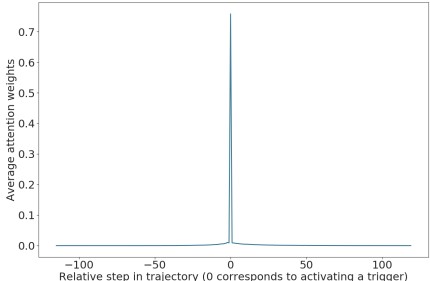 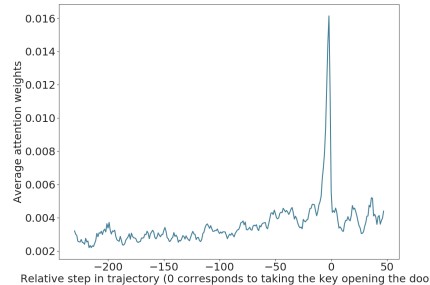

Figure 3: On the **left**, the distribution of attention weights around triggers for correct positive reward predictions in a 8x8 Triggers maze with 3 triggers and one reward. The x-axis denotes the signed number of steps between the state-action couple receiving attention and the closest actual moment the agent activated a switch. On the **right**, the distribution of attention weights around keys for correct reward predictions for door traversals in DMLab.

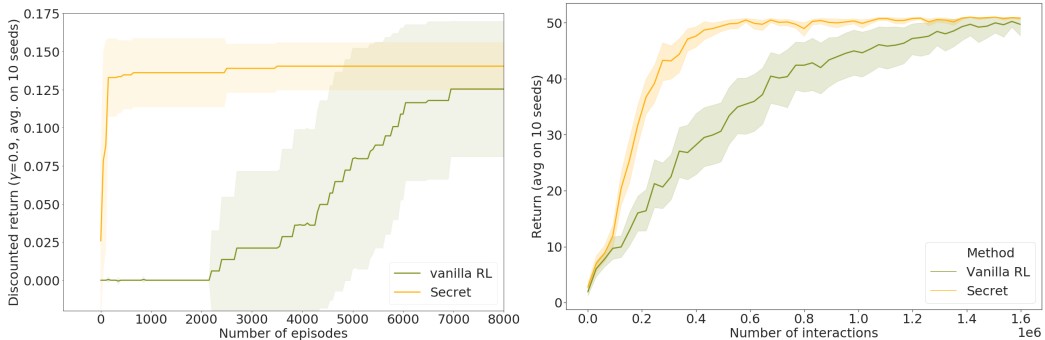

Figure 4: On the **left**, in-domain transfer results on a 8x8 Triggers with 3 triggers and 1 reward. On the **right**, results in DMLab.

## 3.2 TRANSFER

We then study how we can leverage the inferred credit and transfer representations that are helpful in new scenarios. We show that agents train faster when using shaped rewards from SECRET. As before, the reward model is trained on episodes of interaction in environments sampled from the source distribution. In transfer environments, we sample multiple trajectories, each using the same maze configuration. We then compute the attentional potential function by calculating an estimate of the expected redistributed reward, as described in Sec. 2.2. To evaluate its effect, we compare agents trained from environment rewards to agents that use the resulting shaped reward.

**In-domain transfer**   For in-domain transfer, we transfer the representations for credit assignment to held-out instances of the same distribution over MDPs. For the Triggers environment, the RL agents are tabular $Q$-learners. For the DMLab environment, we use PPO agents (Schulman et al., 2017) and modify the original task: we do not reward the agent for collecting keys but only to open doors so that the attention can focus on the key positions. Note that this makes the task harder.

As we display in Fig. 4 agents learn visibly faster to solve tasks when benefitting from SECRET in both environments.

**Out-of-domain transfer**   For out-of-domain transfer we use the Triggers environment and consider two scenarios that are hard for standard agents: transfer to bigger environments (see Fig. 5) and transfer to environments with inverted dynamics (see Fig. 6). In the bigger setting, direct weight transfer cannot be used since the visual input has bigger spatial dimensions. On the other hand, SECRET can be used since the transformation we apply to turn states into observations conserves the visual input dimensions. In the inverted dynamics setting, the effect of the agent's actions are inverted, which makes the task hard for transfer methods. In that setting, we compare the transferability

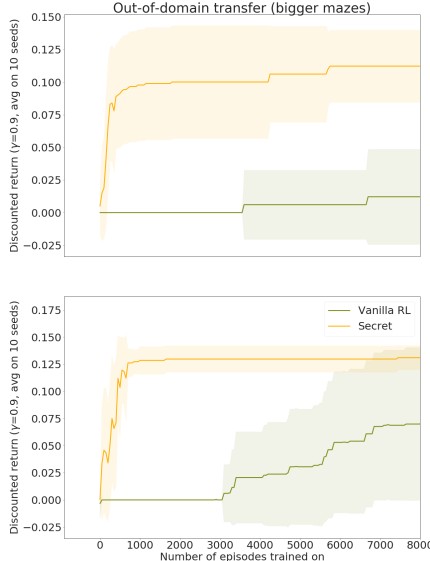

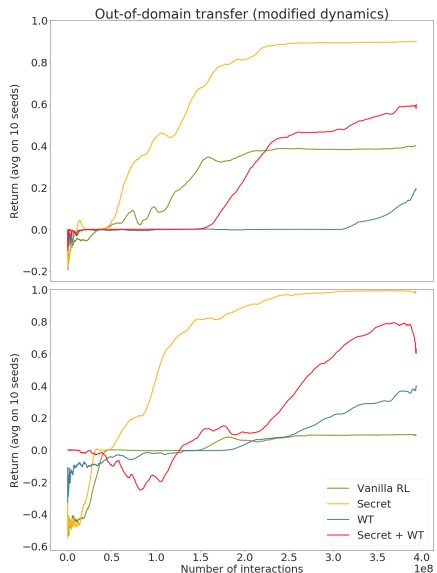

Figure 5: Bigger environments we consider are bigger mazes where the structure of the original task is conserved (number of triggers, number of prizes). Environments drawn are 12x12 grids with 1 trigger and 1 prize for the top figure versus 2 prizes for the bottom one. Environments from the training distribution are 8x8 grids.

Figure 6: The controls of the out-of-domain distribution are inverted (up becomes down, right becomes left). Environments are 8x8 grids with 1 trigger and 1 prize (top figure) or 2 prizes (bottom figure). The effect of the shaping is exclusively beneficial, while transferring weights from the source task can be detrimental to the learning process.

of our mechanism to that of the representations learned by an agent equipped with deep function approximation. To this end we use DQN agents and either train them from scratch in the target environments or start from the set of weights learned in the source environments (WT in Fig. 6).

In both settings, shaping the rewards assists the agent in learning to solve the task. We display some results in Fig. 5 and Fig. 6. When transferring to bigger environments, the agent benefits very early on from the shaped reward, while also reaching better asymptotical performance.

## 4 RELATED WORK

**Transfer in RL** While a lot of approaches exist in the transfer literature, to the best of our knowledge none explicitly transfer credit assignment capabilities. Previous works aimed at making the training of an agent in the same task more sample-efficient by using a pretrained model as a teacher (Rusu et al., 2016a; Schmitt et al., 2018). We learn to assign credit as a parallel task that does not modify the representations of the RL agent. Others learn auxiliary reward functions in the hope that they will enable transfer by imposing consistency in the reward (Houthooft et al., 2018; Hessel et al., 2018; Agarwal et al., 2019). Although we also learn additional reward signal, it is based on a redistribution of rewards from the environment, which ensures consistency with the original reward function. Transfer is also viewed as learning tasks in a sequential way (Rusu et al., 2016b; Kirkpatrick et al., 2017) and this suggests to introduce inductive bias to the neural architectures of agents to reduce catastrophic forgetting. Our method does not require to alter the agent's architecture. Other explicitly address the problem of transfer through the lens of multitask learning (Parisotto et al., 2016; Teh et al., 2017) while we stick to learning from an initial distribution of environments. Meta-learning approaches aim to train agents on a distribution of tasks or environments so that their learned skills and representations work across the underlying continuum, and allow for fast adaptation of the agents (Duan et al., 2016; Wang et al., 2016; Finn et al., 2017; Mishra et al., 2018; Co-Reyes et al., 2019; Zou et al., 2019). In contrast to meta-learning methods, we do not modify the RL algorithm used to train the agent and SECRET is compatible with any core algorithm for RL.

**Credit assignment** Previous works investigated the role of attention mechanisms for credit assignment. Ke et al. (2018) propose SAB, a sparse attention mechanism used to derive a modified backpropagation algorithm. We draw inspiration from SAB but operate in the RL context without sparsity assumptions about the attention weights. RUDDER (Arjona-Medina et al., 2019) proposes to equip RL agents with an online method for credit assignment based on return decomposition. Our method operates offline and decomposes individual rewards. RUDDER requires a specific exploration scheme, an additional episodic replay buffer, a compute-heavy contribution analysis method and the addition of several auxiliary losses to the objective the RL agent optimizes. In comparison, SECRET is a lightweight method that does not deal with exploration. Crucially, the focus of Arjona-Medina et al. (2019) is on online credit assignment while ours is on transfer. Hung et al. (2018) provide an agent with an external memory and the unsupervised task of reconstructing its inputs (both states and rewards). The agent uses memory reads as a way to identify related elements in sequences, and uses those to transfer the value of states providing delayed rewards to the bootstrapping target of contributing elements. In contrast, SECRET makes use of a non-autoregressive architecture, does not reconstruct states, makes use of reward shaping instead of modifying the update function and most importantly does not rely on an external memory. Recall Traces (Goyal et al., 2018) use a generative model that goes backward from high-reward states and samples state-action pairs that could have led to that state. SECRET also works backward from high-reward states but creates links to previous states from existing trajectories instead of sampling them.

## 5 CONCLUSION

In this work, we investigated the role credit assignment could play in transfer learning and came up with SECRET, a novel transfer learning method that takes advantage of the relational properties of self-attention and transfers credit assignment instead of policy weights. We showed that SECRET led to improved sample efficiency in generalization and transfer scenarios in non-trivial gridworlds and a more complex 3D navigational task. To the best of our knowledge, this is the first line of work in the exciting direction of credit assignment for transfer. We think it would be worth exploring how SECRET could be incorporated into online reinforcement learning methods and leave this for future work.

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

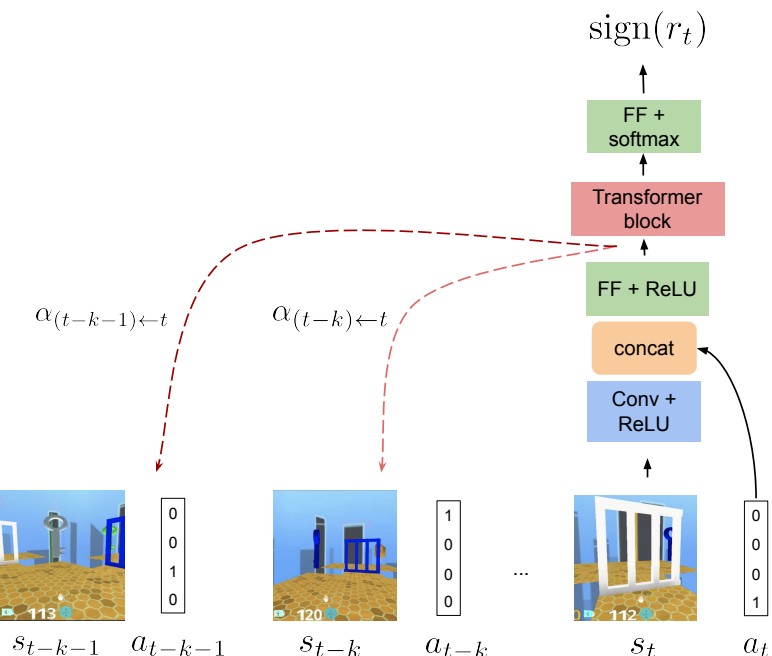

Figure 7: The architecture used for SECRET. $\alpha_{\cdot \leftarrow t}$ is the vector containing the attention weights of the model for its prediction at step $t$.

## A  ADDITIONAL EXPERIMENT DETAILS

In this section we provide additional details about our experimental setup and the hyperparameters we use.

### A.1  TIME LIMITS

In Triggers, episodes stop after 50 timesteps in 8x8 grids and 100 timesteps in 12x12 grids. Those values were chosen large enough such that the policies used to train the reward predictor have a chance to gather informative rewards. In DMLab, episodes stop after 900 timesteps, as defined by the standard settings.

### A.2  REWARD PREDICTION MODEL

We use the same set of hyperparameters in all our experiments with few variation. In Triggers experiments, we use 128 units per dense layer, 32 convolutional filters and a single convolutional layer to process partial states and actions. We apply dropout (Srivastava et al., 2014) at several places in the model as a regularizer. We use a dropout rate of 0.1 after dense layers, a dropout rate of 0.2 in the self-attention mechanism, and a dropout rate of 0.2 in the normalization blocks of the Transformer architecture. Since Transformers create representations via pooling, they need additional information so as to take into account the relative positions of sequence elements. Hence, we use the same positional encoding scheme as in Vaswani et al. (2017), that is we add sine and cosine signals of varying frequencies to the sequence of input embeddings $X_{emb}$ before the self-attention layer:

$$PE_{t,2i} = sin\Big(\frac{t}{\nu^{2i/d_i}}\Big),$$

$$PE_{t,2i+1} = cos\Big(\frac{t}{\nu^{2i/d_i}}\Big),$$

$$X = X_{emb} + PE.$$

In these equations, $t$ refers to the timestep in the sequence, $i$ to the dimension of the embedding considered, and $\nu$ has a constant value of $10000$.

In DMLab experiments, we use two convolutional layers, 16 filters for each, and otherwise identical hyperparameters.

Typically, we set the class weights in the loss function to $w(1) = w(-1) = 0.499$, $w(0) = 0.02$. Fig. 7 gives an overview of the whole architecture.

### A.3 Heuristic for precision-recall analysis

In Sec. 3.1, we compare the attention vectors we get as outputs from the reward prediction model to ideal credit assignment with binary metrics. The ground truth we use is a binary vector of the size of the attention vector. Its values are 0 everywhere and 1 for timesteps that correspond to the activation of a Triggers. To do so, we introduce a simple heuristic to binarize the attention scalars: we consider all values above a threshold $\alpha$ to correspond to events to be credited. Then, we can measure precision and recall as in a binary classification paradigm. The precision and recall reported are the average precision and recall over 4 scenarios : Triggers with a 8x8 grid, 1 trigger and 1 reward; Triggers with a 8x8 grid, 1 trigger and 2 rewards; Triggers with a 8x8 grid, 2 triggers and 2 rewards; and Triggers with a 8x8 grid, 3 triggers and 1 reward. In each scenario, we train the model over a set of $40000$ trajectories, each of which is drawn from a randomly sampled maze. Then, we apply it on $5000$ trajectories from held-out environments and collect the attention weights corresponding to predictions on timesteps where the agent experiences positive reward. We use a fixed $\alpha$ of $0.2$.

### A.4 Transfer in Triggers

We collect $40000$ trajectories sampled from random policies to train the prediction reward model in all distributions of environments.

In experiments involving tabular $Q$-learning we use online $Q$-learning with a learning rate of $0.1$ and a constant greediness factor $\epsilon$ also equal to $0.1$.

For the out-of-domain transfer experiment with modified dynamics, we use a smaller version of the DQN architecture in Mnih et al. (2015). The first convolutional layer has 8 filters, a 3x3 kernel size and a stride of 2. The second and the third convolutional layers have both 16 filters, a 3x3 kernel size and a stride of 1. Those are completed by a feed-forward layer with $64$ units followed by another feedforward layer with as many units as the number of available actions. The greediness factor $\epsilon$ is decayed linearly from 1 to $0.01$ over $250000$ steps in the environment at train time and has a constant value of $0.001$ at test time. We use RMSProp (Tieleman and Hinton, 2012) as an optimizer with a base learning rate of $0.00025$. We update the target network every $2000$ steps and initially fill the replay buffer with $5000$ transitions sampled following a random policy. The replay buffer has a maximum size of $1000000$.

### A.5 In-domain transfer in DMLab

We provide additional details about this setup: we train SECRET using $10000$ trajectories sampled from a distribution of mazes that are generated randomly. These trajectories are sampled using an agent trained over the same distribution. We do so to increase the proportion of trajectories where rewards are experienced. Indeed, we found that using random policies yielded very few of these. Once the model is trained, we use it to compute the attentional potential function over a fixed maze. $1000$ trajectories are sampled on the fixed maze using the same policy as the one that generated the trajectories used to train the reward prediction model. Since consecutive frames can be very similar, we consider a positive reward prediction to be correct (and thus use the corresponding attention weights when estimating the potential) if it happens within 5 frames of a reward actually experienced in the environment. We then compare the performance of agents trained with the original reward function to those trained with the shaped reward.

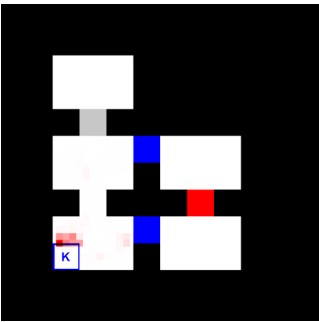

Figure 8: Attention heatmap in DMLab. Attention concentrates around the key which is necessary to unlock reward.

An important point is that we use the knowledge of the position and the keys the agent possesses to build the state used to compute the potential function. This information is not given to the agent. We create an approximate state as such: it is the concatenation of the discretized position and the identifier of the key possessed. The discretized position is the result of the euclidean division by a cell size integer $c_s$ and is necessary since the DMLab position is continuous. $c_s$ is fixed and has the value of 50 in our experiments. We acknowledge that relying on a manually constructed state limits the generality of our approach, but we are confident that this limitation can be addressed in future work by using an estimate of the true state.

All agents mentioned in that section are PPO learners with a learning rate of 0.00019, an entropy coefficient of 0.0011, 12 actors, a discount factor $\gamma = 0.99$. They use generalized advantage estimation (Schulman et al., 2016) with $\lambda = 0.95$. All those hyperparameter values are taken as is from the experiment section in Savinov et al. (2018), whose code is open-source.

### A.6 ATTENTION HEATMAP IN DMLAB

In Fig. 8, we display the per-position average attention weight along trajectories starting in the middle-left room that led to opening the blue doors. Attention (shown in shades of red) concentrates around the key position in the lower-left corner.

## B ADDITIONAL EXPERIMENTS

In this section we provide the results of additional experiments that are useful to understand several aspects of SECRET.

### B.1 INFLUENCE OF THE STATE TRANSFORMATION IN TRIGGERS

We study the influence of the size of the window in the transformation used to turn states into observations in Triggers. The goal of this experiment is notably to study the effect of the degree of partial observability to the assigned credit, and also the effect of transferring a possibly badly assigned credit to an agent, through reward shaping.

We consider the following window types: 3x3, 5x5 and 7x7 (odd numbers because the window is centered on the agent), as well as the full state. We place ourselves in the in-domain setting and evaluate SECRET on held-out environments from the same distribution as the source. Environments are 8x8 mazes with 3 triggers and 1 prize. We display reward prediction and credit assignment metrics for each window size in Table 1, and the average discounted return of tabular $Q$-learning agents for each window size in Fig 9a. Notice that the considered reward prediction accuracy (Table 1) is weighted, such that all classes have the same importance.

In this setting, we can observe that bigger window sizes are detrimental to the quality of the credit assigned, even if it helps to predict reward (Table 1). This was to be expected: the more "observable" is the sequence of observations used to train the reward predictor, the easier is the reward prediction (in the full state case, it can be predicted solely from the current state-action couple), but also the less

Table 1: Influence of the window size on the reward prediction and credit assignment quality.

| Window size | Reward prediction accuracy | Credit precision | Credit recall |
|---|---|---|---|
| 3x3 | $0.67 \pm 0.06$ | $1.0 \pm 0.$ | $1.0 \pm 0.$ |
| 5x5 | $0.92 \pm 0.02$ | $0.57 \pm 0.04$ | $0.38 \pm 0.03$ |
| 7x7 | $0.75 \pm 0.12$ | $0.07 \pm 0.04$ | $0.03 \pm 0.02$ |
| Full state | $0.76 \pm 0.14$ | $0.10 \pm 0.05$ | $0.02 \pm 0.02$ |

Table 2: Influence of the class weighting on the reward prediction and credit assignment quality.

| $w(0)$ | Reward prediction accuracy | Credit precision | Credit recall |
|---|---|---|---|
| 1.0 | $0.72 \pm 0.15$ | $0.83 \pm 0.34$ | $0.59 \pm 0.27$ |
| 0.1 | $0.72 \pm 0.09$ | $1.0 \pm 0.$ | $0.89 \pm 0.05$ |
| 0.01 | $0.68 \pm 0.09$ | $1.0 \pm 0.$ | $0.93 \pm 0.05$ |
| 0.001 | $0.65 \pm 0.04$ | $1.0 \pm 0.$ | $0.95 \pm 0.03$ |

likely the assigned credit will be centered on the trigger. However, even with a low credit precision, the shaped reward still accelerates the learning process (Fig. 9a).

## B.2 INFLUENCE OF THE CLASS WEIGHTS IN THE REWARD PREDICTION LOSS

We stated that class weighting was important to keep the variance of prediction metrics across a variety of datasets of sampled trajectories low, the classes (sign of the reward) being quite imbalanced. Here, we study the influence of the class weights considered in the loss of the reward prediction model in Triggers.

We consider the following class weights for the class corresponding to zero rewards: $w(0) = 1$, $w(0) = 0.1$, $w(0) = 0.01$ and $w(0) = 0.001$. The other class weights are fixed and have the value $w(1) = w(-1) = 1$. We place ourselves in the in-domain setting and evaluate SECRET on held-out environments from the same distribution as the source. Environments are 8x8 mazes with 3 triggers and 1 prize.

We display reward prediction and credit assignment metrics for each window size in Table 2, and the average discounted return of tabular $Q$-learning agents for each window size in Fig 9b. In that setting, putting heavier misclassification penalties for under-represented classes ($-1$ and $1$) results in improved credit and RL performance.

## B.3 INFLUENCE OF THE AMOUNT OF AVAILABLE DATA

We study the influence of the amount of data used by SECRET in a Triggers task. The objective is to assess how data-demanding is the proposed approach, and the effect of a lack of data on the transfer. There are two types of data: the data from the source domain (used to train its reward prediction component), and the data from the target domain (used to build the potential function).

### B.3.1 SOURCE DISTRIBUTION

We consider the following number of trajectories to train the self-attentive reward predictor: 500, 1000, 5000, 10000, 25000 and 50000. We place ourselves in the in-domain setting and evaluate SECRET on held-out environments from the same distribution as the source. Environments are 8x8 mazes with 3 triggers and 1 prize.

We display reward prediction and credit assignment metrics for each window size in Table 3, and the average discounted return of tabular $Q$-learning agents for each number of trajectories in Fig. 9c. We observe that increasing the size of the dataset improves the results, both for the efficiency of the reward predictor (Table 3) and for the transfer (Fig. 9c). Having too few data can slow down learning compared to a vanilla agent (Fig. 9c), but it does not prevent from learning to solve the task.

Table 3: Influence of the number of episodes in the source domain on the reward prediction and credit assignment quality.

| Number of source episodes | Reward prediction accuracy | Credit precision | Credit recall |
|---|---|---|---|
| 500 | $0.53 \pm 0.11$ | $0.23 \pm 0.37$ | $0.09 \pm 0.2$ |
| 1000 | $0.65 \pm 0.07$ | $0.49 \pm 0.33$ | $0.52 \pm 0.42$ |
| 5000 | $0.79 \pm 0.08$ | $0.9 \pm 0.30$ | $0.9 \pm 0.30$ |
| 10000 | $0.84 \pm 0.07$ | $0.9 \pm 0.30$ | $0.9 \pm 0.30$ |
| 25000 | $0.88 \pm 0.09$ | $0.9 \pm 0.30$ | $0.9 \pm 0.30$ |
| 50000 | $0.83 \pm 0.11$ | $0.8 \pm 0.4$ | $0.8 \pm 0.4$ |

### B.3.2 TARGET DISTRIBUTION

We consider the following number of trajectories sampled from the target environment to build the potential function: 100, 500, 1000, 2000, 5000, 10000. We place ourselves in the in-domain setting and evaluate SECRET on held-out environments from the same distribution as the source. Environments are 8x8 mazes with 3 triggers and 1 prize.

We display the average discounted return of tabular $Q$-learning agents for each number of trajectories in the source domain in Fig.9d. Here again, we observe that the number of episodes in the target distribution is an important parameter for SECRET, but it has less impact than the number of episodes in the source domain.

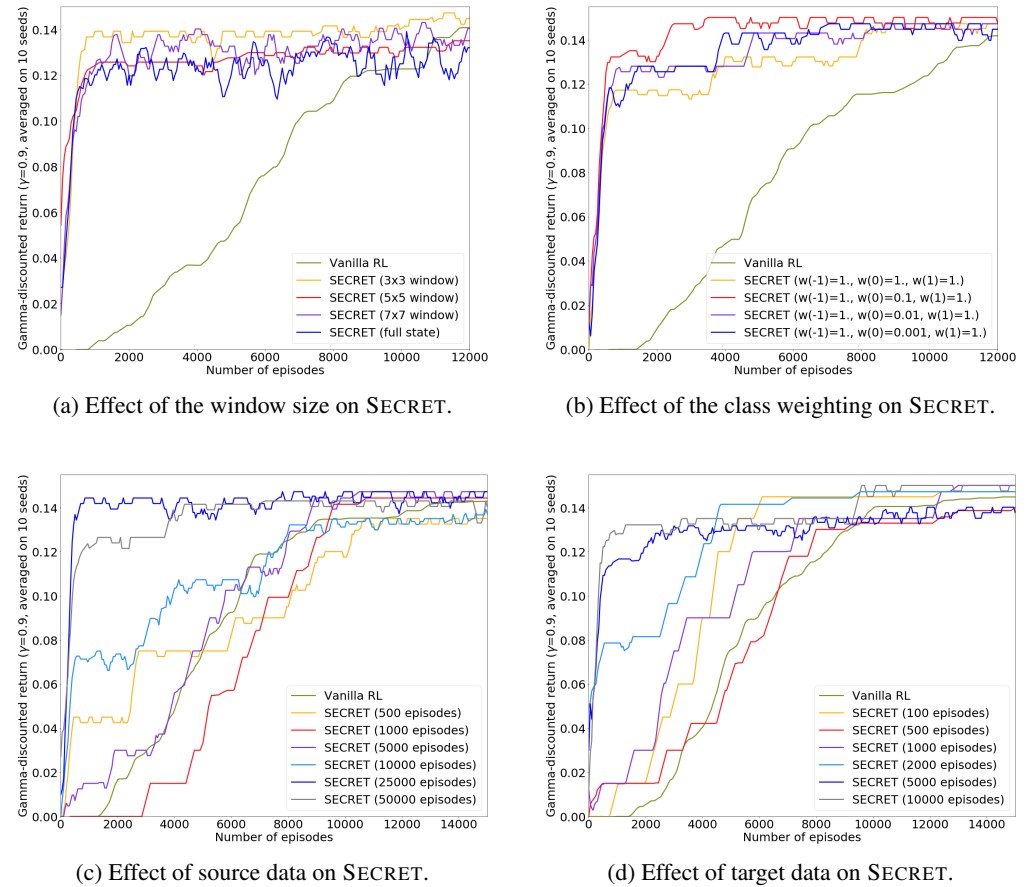

(a) Effect of the window size on SECRET.

(b) Effect of the class weighting on SECRET.

(c) Effect of source data on SECRET.

(d) Effect of target data on SECRET.

Figure 9: Effect of various parameters on SECRET.

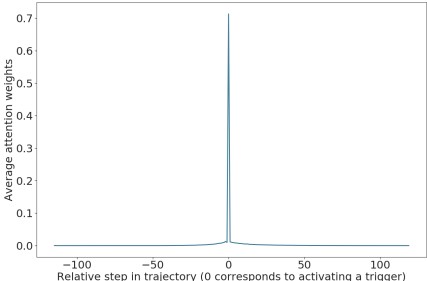 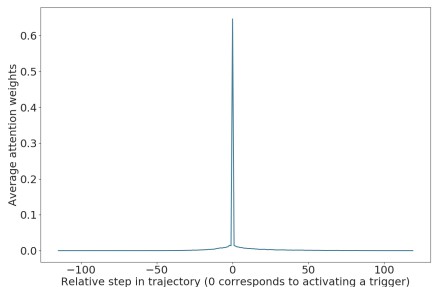

Figure 10: On the **left**, the distribution of attention weights around triggers for correct positive reward predictions in a Triggers maze from an out-of-domain distribution (bigger mazes). The x-axis denotes the signed number of steps between the state-action couple receiving attention and the closest actual moment the agent activated a switch. On the **right**, same figure for another out-of-domain distribution (inverted dynamics).

## B.4 ATTENTION DISTRIBUTIONS IN OUT-OF-DOMAIN SCENARIOS

In the main paper, we have only shown the distribution of attention for an in-domain scenario. Here, following the same protocol as in Sec. 3.1, we measure the distribution of attention weights in the two out-of-domain scenarios considered in our experiments: bigger mazes and inverted dynamics. The results are reported in Fig. 10. We observe a similar distribution of attention, peaked on the triggers.

