# OpenReview forum: "Self-Attentional Credit Assignment for Transfer in Reinforcement Learning"
_ICLR.cc/2020/Conference — Reject_

### Official Review · AnonReviewer3 · 2019-10-23
**Official Blind Review #3**

**Rating:** 3

**Review:**

This work focuses on credit assignment using a self-attention module for transfer RL problems. Specifically, the reward signals are assigned backward to previous states according to the attention weights. This can be helpful especially when the reward signal is sparse. Experiments on the newly proposed Triggers environment and the DMLlab keys & doors environment show that the proposed algorithm, SECRET, can speed up training in the transferred environment.

Pros:
- The writing is mostly great.

Cons:
- Some design choices are not well-motivated or even problematic.
- Experiments are not sufficient.

(1) On page 3, the definition of Z is problematic. The mask matrix M is applied *before* the softmax transformation, which means future values can have non-zero attention. This is because softmax will never produce zero probability. It would still be problematic even if M is applied after the softmax transformation because in this case, the attention for past elements could become very small and almost surely not sum to 1 (except for the last element). Therefore, regardless of the position of M, the attention will be questionable.

(2) Page 4, the weight w(c) is not defined for the weighted cross-entropy. It is claimed that such weighting is essential, but no evidence is provided to support this.

(3) The proposed potential function is not very well-motivated. It is not clear why it should be defined like this instead of other alternatives. Moreover, for never-visited states, the potential is set to 0, which seems to prevent exploration. This would potentially harm the performance in a new environment especially when the training trajectories are far from optimal.

(4) Sec.2.3 says "given an underlying environment state and a specific action, their contribution to future rewards is not fundamentally altered." Can you elaborate? Also what is "the rank of the rewards"?

(5) Experiment:
(5.1) Why Fig.5 and 6 do not have similar asymptotic returns? Given that they both correspond to 1 trigger and 1 (2) prize(s), the asymptotic return should be close.
(5.2) As mentioned above, it would be interesting to see whether SECRET will prevent exploration if the behavior agent is (heavily) biased. The random agent in the Triggers environment provides sufficient support for the whole state space, while the PPO agent in the DMLab is well-focused on the "good" regions. If a "bad" agent (say, exploits some low reward regions) is used, SECRET may slow down instead of speed up training in the transfer environment. This is an important scenario to see whether SECRET will potentially create a negative transfer.
(5.3) No other method from the literature is used for comparison. Several alternatives are discussed in the Related Work "credit assignment" section, but none is compared in the experiment.

Minor comments
- In other fields than RL -> in fields other than RL.
- The caption of Fig.3-left: there is no "key" in the Triggers environment. It uses switches.
- WT is not explained in Fig.6.
- h(s) is defined as the observation given a state s, but it is not used in later discussion.

**Experience Assessment:**

I have read many papers in this area.

**Review Assessment: Checking Correctness Of Derivations And Theory:**

I assessed the sensibility of the derivations and theory.

**Review Assessment: Checking Correctness Of Experiments:**

I assessed the sensibility of the experiments.

**Review Assessment: Thoroughness In Paper Reading:**

I read the paper at least twice and used my best judgement in assessing the paper.

---

> ### Author Response · Authors · 2019-11-15
> **Re: Official Blind Review #3 (1/2)**
>
> 1. We agree with the line of reasoning of the reviewer and thank them for their careful checking. There is indeed a mistake in the equation defining Z. The intended effect of the causal mask is either to conserve values or assign a negative number with a large magnitude so that the future values are set to zero after the softmax operation. In our experiments, this is exactly what is being done: the causal mask multiplies elementwise the key-query product matrix and the result of that operation is added with the negation of the mask multiplied by a negative constant with a large magnitude. Thus, the problem only lay in the equation mentioned, which did not correspond to the reality of the experiments. We updated in the equation in the draft accordingly.
>
> 2. The values of the weights w(c) are defined in Appendix A.1. The rationale behind using class weights in the sequential cross-entropy is that the reward prediction task is highly imbalanced in the tasks considered. Indeed, agents experience zero rewards most of the time. Hence, we opted for a change in the loss that puts a greater cost on the misclassification of nonzero rewards.
> In the new Appendix B.2. we included an additional experience in which we measure the impact of the class weights values on SECRET in a Triggers scenario. Briefly, we observe that not compensating for class imbalance degrades results.
>
>
> 3. A good way of reducing the detrimental effect of delayed rewards is to assign the reward directly to the responsible actions (as demonstrated in [1]), which is what we aim to do with SECRET. Constructing the potential function from the attention mechanism seems natural to us since we use attention to identify the contributions to future reward. There are certainly other options but this is the one we chose to explore and it is supported by our experimental analysis. Additionally, in theory, since we use reward shaping, it does not modify the set of optimal policies and agents will still explore as they learn.
> Another approach would be to use a stochastic estimate to modify the reward function over a single trajectory, so that we do not have to generate trajectories in the target domain and so that we do not use a potential function. We leave the investigation of that option for future work.
>
> 4. By this sentence ("given an underlying environment state and action, their contribution to future rewards is not fundamentally altered") we mean that if the “structure” of the MDP is preserved, then the credit to be assigned is also preserved. For instance, in the Triggers example, whatever the environment layout or size, the action of activating a trigger contributes the same way to future reward. Of course, depending on the layout, activating a trigger from a given state could mean going up or left, but we believe that the network learns this higher level semantic of "activating a trigger" from episodes in the source domain. Indeed, the representations of actions and observations get mixed in the proposed architecture, making it realistic to learn a combined abstraction.
>
> As for the second part of the question, we agree that the formulation is confusing. What we meant is to consider changes to the reward function that preserve the set of optimal policies (e.g. in Triggers adding a negative reward to the trigger, but not too low such that the optimal policy still consists in exiting the room), and it is potentially more general than transformations that preserve the ranking of the individual values the reward function can take. We changed our formulation.

---

> > ### Author Response · Authors · 2019-11-15
> > **Re: Official Blind Review #3 (2/2)**
> >
> > 5. (1.) Fig.5 reports the discounted return (with gamma = 0.9) while Fig.6 reports the undiscounted return, which explains the different asymptotical values.
> >
> > (2.) Given the properties of reward shaping, we think that in the worst case scenario SECRET should only slow down the learning process of the agent. Also, please notice that we use the attention provided by the reward predictor only if the non-negative reward is correctly predicted (this was not clear initially, we clarified it). This way, we ignore attention that lead to badly predicted reward, which should mitigate the raised problem.
> >
> > As an experimental illustration, we run SECRET with increasing size of the window around the agent in Triggers (as suggested by R2). Increasing the size leads to an inaccurate matching between attention and triggers, as reported in Appendix B.1. We trained agents using the shaped reward obtained from these reward predictors. Results vary depending on the size of the window, but SECRET still helps compared to the vanilla RL (no reward shaping using attention). Related results are provided in the new Appendix B.1.
> >
> > (3.) We propose a method using credit assignment in an offline manner to achieve transfer in RL. SECRET learns representations for credit assignment that are kept separate from those learned in the RL task, which we think is key for transferability. Existing credit assignment methods work online and in combination with an RL agent, which makes them hardly comparable to SECRET. Studying their transferability is outside the scope of the paper (please also see our reply to point 1 of R2, which is related). However, we compare the performance of our method to a baseline, but a transfer baseline, namely based on the transfer of weights.
> >
> > Minor comments
> > WT refers to weight transfer (in which we transfer the weights of an RL agent trained over episodes drawn from the source domain), this is now clarified.
> > We took their comments into account and modified the draft accordingly.
> >
> > [1] Arjona-Medina J., Gillhofer M., Widrich M., Unterthiner T., Brandstetter J., Hochreiter S. - RUDDER: Return Decomposition for Delayed Rewards. NeurIPS 2019.

---

### Official Review · AnonReviewer2 · 2019-10-24
**Official Blind Review #2**

**Rating:** 6

**Review:**

This paper proposes a novel transfer learning mechanism through credit assignment, in which an offline supervised reward prediction model is learned from previously-generated trajectories, and is used to reshape the reward of the target task. The paper introduces an interesting new direction in transfer learning for reinforcement learning, that is robust to the differences in the environtment dynamics.

I have the following questions/concerns.

1. The authors insist that their fous is on transfer and not competing on credit assignment. If accurate credit assignment leads to better transfer, shouldn't achieving the best credit assignment model (thus competing in credit assignment) lead to better transfer results?

2. What effect does the window size for transforming states to observations have on the performance of SECRET?

3. On a high-level, how does SECRET compare to transfer through relational deep reinforcement learning: https://arxiv.org/abs/1806.01830? Relational models use self-attention mechanisms to extract and exploit relations between entities in the scenes for better generalization and transfer. Although SECRET intentionally avoids using relations, I think a discussion around relational models for RL is warranted. I'm curious what happens if SECRET is allowed to exploit relations in the environment.

4. What happens if the reward model uses very few trajectories and is not able to predict good rewards? Does transfer through credit assignment become detrimental? In other words, in a real-world scenario, how I do know when to start using SECRET, or when am I better off learning from environment rewards alone? Especially given that SECRET requires 40000 trajectories in the source domain.

5. Are the samples generated in the target domain for collecting attention weights included in the number of episodes when evaluating SECRET? For example, in Figure 4. I believe the number of episodes required to collect those target samples should be added to the number of episodes when using SECRET since the agent must interact with the environment in the target domain.

6. On a lighter note, I don't believe using a coffe-brewing machine has a 'universally invariant structure' of coffee-making. That's a luxurious way of making coffee :) In the developing world, we still need to boil water, pour coffee powder in it, etc., all without a coffee-brewing machine.

**Experience Assessment:**

I have read many papers in this area.

**Review Assessment: Checking Correctness Of Derivations And Theory:**

I carefully checked the derivations and theory.

**Review Assessment: Checking Correctness Of Experiments:**

I carefully checked the experiments.

**Review Assessment: Thoroughness In Paper Reading:**

I read the paper at least twice and used my best judgement in assessing the paper.

---

> ### Author Response · Authors · 2019-11-15
> **Re: Official Blind Review #2 (1/1)**
>
> 1. We believe that the transferability of SECRET is due to two major aspects: 1) that we keep representations for the credit assignment separate from those for the RL task and 2) that we use a self-attentional architecture, which was shown to transfer in settings other than RL.
> Better credit assignment is desirable and should arguably lead to better transfer results in the case of SECRET. Nevertheless, it is not necessarily true for other credit assignment methods available because they are designed for the online setting and intricately coupled with an RL agent.
> The focus of the paper being on transfer, we proposed a transfer method relying on credit assignment. In our opinion, comparing its credit assignment capabilities to other existing methods is outside of the scope of the paper.
>
> 2. We included the results of varying the window size in the new Appendix B.1. Briefly, with bigger windows, there is less partial observability, and the attention no longer matches the trigger. Please see the new appendix for more details.
>
> 3. Relational Deep RL ([1]) uses spatial self-attention to infer and leverage relations between "objects" (pixel representations). Crucially, it does not make use of the sequential aspect of the RL task. Instead, SECRET relies on temporal credit assignment, which could be presented as a form of temporal relations (as dictated by the reward function). Those are very different approaches to handling relations (if SECRET can be deemed as relational). We think it would indeed be an interesting research direction to combine both spatial and temporal aspects for credit assignment or relational reasoning.
>
> 4. There are two different aspects here: 1) the reward model could be trained on very few trajectories in the source domain, or 2) it could be applied on very few trajectories to build the potential function in the target domain.
> For 1), in practice, we only redistribute the nonzero rewards that were successfully predicted by the reward model, so insufficient prediction capabilities are not a problem. We added a sentence in the main text to mention the fact that we consider correctly predicted nonzero rewards. If the model does not manage to predict nonzero rewards, then SECRET falls back to the Vanilla RL case. In the worst case scenario, SECRET could predict a small proportion of the nonzero rewards and assign wrong credit, which could lead to a slowed down procedure.
> For 2), the potential function used in SECRET relies on trajectories with nonzero rewards. In the worst case scenario, the potential function could not reflect accurately the structure of the MDP and lead to a slowed down learning procedure.
> We now include two additional experiments in Appendix B.3 that explore both scenarios. We show that with a small number of trajectories, either in the source or the target domain, the performance of the agent does not drop too much.
>
> 5. The samples generated in the target domain are not included in the number of episodes reported in the paper. While debatable, our motivation to do so is that we use the same fixed policy we used in the source domain to generate those trajectories. Note that there is no learning procedure involved during the collection of the target samples.
>
> 6. Maybe a follow-up to consider for the coffee test is to adapt from using a coffee-brewing machine to making it from scratch :)
>
> [1] Zambaldi V., Raposo D., Santoro A., Bapst V., Li Y., Babuschkin I., Tuyls K., Reichert D., Lillicrap T., Lockhart E., Shanahan M., Langston V., Pascanu R., Botvinick M., Vinyals O., Battaglia P. - Deep Reinforcement Learning with Relational Inductive Biases. ICLR 2019.

---

### Official Review · AnonReviewer4 · 2019-11-01
**Official Blind Review #4**

**Rating:** 8

**Review:**

This paper proposes to consider the problem of transfer in the context of sequential decision-making -- in particular reinforcement learning -- from the view-point of learning transferable credit assignment capability. They hypothesize that by learning how to assign credit, structural invariants can be learned which the agent can exploit to assign credit effectively and thus learn more efficiently in new environments (be it in-domain or out-of-domain). They pose the credit assignment problem as learning to predict (sparse) rewards at the end of sub-trajectories, finding the extent to which past state-action pairs appear to be responsible for these rewards (by means of the reward-prediction training), and creating a dense reward function via reward shaping (such that the set of optimal policies does not change). This is appealing as no modifications are needed to the RL algorithm/architecture. To examine their hypothesis, they created a method, called SECRET, based on self-attention and supervised learning to train credit assignment capability offline: sample many trajectories (often a mixture of expert and non-expert ones) from the source distribution, train a self-attentive seq2seq model to predict the rewards in these trajectories offline. Once this model is trained, they apply this model to a relatively small set of sampled trajectories from the target distribution and obtain the attention weights. Then, they use these attention weights as a proxy for credit assignment and, thus, use them to form a reward redistribution function. In their experiments, they show that the average attention weights actually signal the state-actions at which the future reward is triggered. They also show in their experiments that SECRET improves learning performance on in-domain transfer learning (larger mazes), as well as an out-of-domain case (with modified dynamics).

Overall, this paper proposes an interesting general avenue for research in transfer learning in RL. Regarding the proof-of-concept method and experiments, I need some clarifications. Given these clarifications in the authors' response, I would be willing to increase my score.

1. Regarding this statement on breaking Markov property: "hide a certain amount of information from states and break the Markov assumption".
(i) It is unclear to me what this "certain amount" would need to be in general. I believe this would require domain-specific knowledge to know what can be removed to break Markov-ness while not introducing significant state-aliasing (which could hinder the agent's learning).
(ii) Does any extent of partial-observability warrant that the success in reward-prediction would mean that we have a valid credit assignment model? I feel like this is not generally true, in which case I question the statement on p.3: "Note that in POMDPs, this is unnecessary since the observations agents get from the environment are incomplete by construction."
(iii) Regarding generality, the fact that states need to be (manually) preprocessed seems to me like a downside of this approach. Can you see any way around this?

2. In p.4, this is mentioned: "In POMDPs, we recover an approximate state from the observation...".
I do not see how this is done in the DMLab experiments. If this is done manually, and not trained, then I think it should be clearly stated in the main text. I think the 2nd paragraph of Sec. A.4 is stating that extra information about the state was utilized, and not approximated via a trained model to recover the states (i.e., no s^=h^-1(o) was used)?

3. What is the observation type of Vanilla RL in the out-of-domain experiments? Is it also observing its local-view (similar partial observability as SECRET) or does it have access to the full state? I would argue that it is important that the performance of Vanilla RL with partial observation is reported. Including both cases could also be beneficial.

4. Fig.3 shows attention weights on held-out environments from an identical distribution as the source (i.e., in-domain).
I would like to see how well the attention signal works when the target distribution differs from the source. Is there a reason why this is not demonstrated?

5. Not sure about specific definitions of sub-trajectory and trajectory in the paper:
(i) What constitutes a sub-trajectory (as opposed to a trajectory) in the context of this paper?
(ii) Are the lengths of the sub-trajectories or trajectories fixed?

6. Why do the attention weights not sum to 1 in Fig.3?

7. Could you clarify the role of positional encoding and how it is done?


Minor comments:

1. M is used to denote both MDP and causal map.
2. Explicitly defining d_i in p.3 should improve clarity.
3. Using 40k and 10k trajectories of interactions to train the credit-assignment model (on Triggers and DMLab domains, respectively) seems quite demanding, which seems somewhat unrealistic to deem useful for application to robotics perhaps?

**Experience Assessment:**

I have read many papers in this area.

**Review Assessment: Checking Correctness Of Derivations And Theory:**

I carefully checked the derivations and theory.

**Review Assessment: Checking Correctness Of Experiments:**

I carefully checked the experiments.

**Review Assessment: Thoroughness In Paper Reading:**

I read the paper thoroughly.

---

> ### Author Response · Authors · 2019-11-15
> **Re: Official Blind Review #4 (1/2)**
>
> 1. (i) We do not currently have a procedure to determine the amount of information to be hidden from states. We acknowledge that we currently need to design the transformation in a task-specific manner, which can be natural (eg, for a robotic task, it could be removing the information about velocity and acceleration) or not. In Triggers, the transformation we consider (cropping the frame around the agent) is natural and could work more generally for navigation tasks.
> We added an experiment where we study the effect of varying the window size used in the transformation applied to Triggers states in the new Appendix B.1. Results show that it is an important parameter for SECRET to assign sensible credit, but also that SECRET speeds up the learning procedure even with little or no partial observability.
> Finally, transformations could indeed lead to state-aliasing (in the sense that they could have the same output for different inputs), but will not affect the agent directly since the agent uses the full state as in standard RL methods, if available. Hence, we believe state-aliasing to be a threat only if states that are crucial for future rewards have identical outputs than other unrelated states.
>
> (ii) We think the point of the reviewer is fair and that partial observability alone might not be sufficient. This is supported by the poor credit assignment quality when using a 7x7 window transformation in Triggers (see the new Appendix B.1).
> With SECRET, valid credit assignment relies on isolating the information necessary to predict that there is reward to be experienced from the corresponding inputs. In our experiments, we find that when using the transformation, architecture and loss we propose for SECRET there is a high correlation between the reward prediction quality (as measured by weighted accuracy) and the quality of the credit assigned (measured by the precision and recall calculated as in Sec.3.1). We only have empirical evidence of this for now, and no general theoretical result.
> We modified the sentence the reviewer mentioned to reflect these points.
>
> (iii) The preprocessing we do is rather natural for the considered problem, and maybe more generally for a navigation task, but we acknowledge that it is not general. We can think of some options that might alleviate the need for manual preprocessing:
> We could use a reconstructed version of the input. For instance, we could apply noisy autoencoding on the state representation of the RL agent and use the result as input to the reward prediction model.
> In [1] they use auxiliary losses to have their model attend to past sequence elements despite having access to the current state.
> We could hide a subset of sequence elements so that the model has incentive to diversify its focus. The subset could be reduced to the current sequence element or be determined randomly.
> We leave the exploration of automatic preprocessing strategies for future work, other approaches could be envisioned.
>
> 2. In our DMLab experiment, we indeed use the position and the current key possessed by the agent to infer an imperfect state that we use to create the potential function.
> The state is created as such: it is the concatenation of the discretized position and the identifier of the key possessed. The discretized position is the result of the euclidean division by a cell size integer c_s and is necessary since the DMLab position is continuous. c_s is fixed and has the value of 50 in our experiments.
> We modified the paper so that the fact we manually construct the state is clear from the main text, and that the method used to create the state is clear from the Appendix.
>
> 3. In Triggers (and more generally in MDPs), the observation type of vanilla RL in both in-domain and out-of-domain experiments is the full state. The observation built from the state is only used as input to the sequence model tasked with reward prediction. We did not consider using partial observations as input to RL agents since we think it would make the task artificially harder and would potentially require a recurrent architecture for the agents.
>
> In DMLab, the observation type in all tasks and for both agents and the sequence model is the partial observation.
>
> 4. We added attention distributions in out-of-domain scenarios in the new Appendix B.4.
> Note that they look nearly identical to the updated attention distribution in the in-domain case, which is backed by near perfect precision and recall metrics.

---

> > ### Author Response · Authors · 2019-11-15
> > **Re: Official Blind Review #4 (2/2)**
> >
> > 5. (i) A trajectory is a sequence of observation-action pairs that covers a whole episode of interaction. In the context of our paper, a sub-trajectory is a portion of trajectory that starts at the beginning of the episode and is now defined as such in the main text. The reward prediction task considers all sub-trajectories. Another way to look at it is that for each trajectory in the dataset, we try to reconstruct whole sequences of rewards from the observation-action pairs.
> >
> > (ii) The lengths of trajectories is not fixed. However, in our experiments, they are upper bounded due to time limits in Triggers and DMLab (in Triggers, the time limit depends on the size of the grid and is chosen so that it is sufficient to observe rewards given the played policies, while in DMLab it is fixed to 900).
> > We added information about the time limits used in Appendix A.
> >
> > 6. Attention weights sum to 1 in our experiments. The distribution displayed on Fig.3 mistakenly reports normalized logits instead of post-softmax weights, which explains why the values do not sum to 1. We updated the figures in the paper so that the post-softmax values are reported instead of normalized logits.
> > Note that the updated average attention weights sum to 1 and also that the attention remains peaked around interest points in the updated distributions. The improvement over the previous figure lies in the application of the softmax and the use of class weights in the loss.
> >
> > 7. Since Transformers create representations via pooling, they need additional information so as to take into account the relative positions of sequence elements. Positional encoding is a way to incorporate that, and it consists in adding sine and cosine signals of varying frequencies to the input embeddings of the Transformer. Positional encoding is a building block of Transformer models that is well described in [2].
> > We added a description of the effect of positional encoding in the main text and describe how it is done in practice in Appendix A.
> >
> > Minor comments
> >
> > 1- We now use M_c as the notation for the causal mask.
> > 2- We now state that d_i refers to the dimensionality of inputs in the main text.
> > 3- The demand in the number of trajectories for Triggers is high due to the inefficiency of the random policy. For instance, in a 8x8 Triggers environment with 3 triggers and 1 prize only 1.6% of the trajectories feature a positive reward. Using suboptimal trajectories (from an imperfect learner or backup trajectories from past experimentation) could alleviate this demand.
> > Though, we agree that further experimentation would be needed to establish whether the proposed method is suitable for robotics.
> >
> > [1] Arjona-Medina J., Gillhofer M., Widrich M., Unterthiner T., Brandstetter J., Hochreiter S. - RUDDER: Return Decomposition for Delayed Rewards. NeurIPS 2019.
> > [2] Vaswani A., Shazeer N., Parmar N., Uszkoreit J., Jones L., Gomez A., Kaiser L., and Polosukhin I. - Attention is all you Need. NeurIPS 2017.

---

### Public Comment · ~Su_Young_Lee1 · 2019-09-30
**A question related to the attention weights in Figure 3**

I really enjoyed reading this paper especially the novel idea to apply credit assignment in the transfer learning domain.

I have a question regarding on the attention weights reported in Figure 3.
As I understand, the attention weights are generated from a vector of softmax attention, therefore should sum up to 1.
It seems like the sums of the average attention weights in Figure 3 (both left and right) are  far above 1.

I would appreciate if you could let me know whether I missed or misunderstood some experimental settings.
Thank you!

---

> ### Author Response · Authors · 2019-09-30
> **Re: a question related to the attention weights in Figure 3**
>
> Thanks for the comment.
>
> Attention weights indeed sum to 1. The distribution displayed on Figure 3 mistakenly reports normalized logits instead of post-softmax weights, which explains why the values do not sum to 1. Note that the attention remains peaked around interest points in the updated distributions.
>
> We will rectify in the final draft.

---

### Decision · Program_Chairs · 2019-12-19

**Decision:**

Reject

**Comment:**

The paper introduces a novel approach to transfer learning in RL based on credit assignment. The reviewers had quite diverse opinions on this paper. The strength of the paper is that it introduces an interesting new direction for transfer learning in RL. However, there are some questions regarding design choices and whether the experiments sufficiently validate the idea (i.e., the sensitivity to hyperparameters is a  question that is not sufficiently addressed). Overall, this research has great potential. However, a more extensive empirical study is necessary before it can be accepted.